# The relationship between workload and burnout among nurses: The buffering role of personal, social and organisational resources

Elisabeth Diehl[1], Sandra Rieger[1], Stephan Letzel[1], Anja Schablon[2], Albert Nienhaus[2,3], Luis Carlos Escobar Pinzon[1,4‡], Pavel Dietz[1‡]*

1 Institute of Occupational, Social and Environmental Medicine, University Medical Center of the Johannes Gutenberg University Mainz, Mainz, Germany, 2 Institute for Health Services Research in Dermatology and Nursing (IVDP), University Medical Center Hamburg-Eppendorf, Hamburg, Germany, 3 Department for Occupational Medicine, Hazardous Substances and Health Science, Institution for Accident Insurance and Prevention in the Health and Welfare Services (BGW), Hamburg, Germany, 4 Federal Institute for Occupational Safety and Health (BAuA), Berlin, Germany

‡ These authors are joint senior authors on this work.
* pdietz@uni-mainz.de

**Data Availability Statement:** According to the Ethics Committee of the Medical Association of Rhineland-Palatinate (Study ID: 837.326.16

## Abstract

Workload in the nursing profession is high, which is associated with poor health. Thus, it is important to get a proper understanding of the working situation and to analyse factors which might be able to mitigate the negative effects of such a high workload. In Germany, many people with serious or life-threatening illnesses are treated in non-specialized palliative care settings such as nursing homes, hospitals and outpatient care. The purpose of the present study was to investigate the buffering role of resources on the relationship between workload and burnout among nurses. A nationwide cross-sectional survey was applied. The questionnaire included parts of the Copenhagen Psychosocial Questionnaire (COPSOQ) (scale 'quantitative demands' measuring workload, scale 'burnout', various scales to resources), the resilience questionnaire RS-13 and single self-developed questions. Bivariate and moderator analyses were performed. Palliative care aspects, such as the 'extent of palliative care', were incorporated to the analyses as covariates. 497 nurses participated. Nurses who reported 'workplace commitment', a 'good working team' and 'recognition from supervisor' conveyed a weaker association between 'quantitative demands' and 'burnout' than those who did not. On average, nurses spend 20% of their working time with palliative care. Spending more time than this was associated with 'burnout'. The results of our study imply a buffering role of different resources on burnout. Additionally, the study reveals that the 'extent of palliative care' may have an impact on nurse burnout, and should be considered in future studies.

(10645)), the Institute of Occupational, Social and Environmental Medicine of the University Medical Center of the University Mainz is specified as data holding organization. The institution is not allowed to share the data publically in order to guarantee anonymity to the institutions that participated in the survey because some institution-specific information could be linked to specific institutions. The data set of the present study is stored on the institution server at the University Medical Centre of the University of Mainz and can be requested for scientific purposes via the institution office. This ensures that data will be accessible even if the authors of the present paper change affiliation. Postal address: University Medical Center of the University of Mainz, Institute of Occupational, Social and Environmental Medicine, Obere Zahlbacher Str. 67, D-55131 Mainz. Email address: arbeitsmedizin@uni-mainz.de.

**Funding:** The research was funded by the BGW - Berufsgenossenschaft für Gesundheitsdienst und Wohlfahrtspflege (Institution for Statutory Accident Insurance and Prevention in Health and Welfare Services). The funders had no role in study design, data collection and analysis, decision to publish, or preparation of the manuscript.

**Competing interests:** I have read the journal's policy and the authors of this manuscript have the following competing interests: The project was funded by the BGW - Berufsgenossenschaft für Gesundheitsdienst und Wohlfahrtspflege (Institution for Statutory Accident Insurance and Prevention in Health and Welfare Services). The BGW is responsible for the health concerns of the target group investigated in the present study, namely nurses. Prof. Dr. A. Nienhaus is head of the Department for Occupational Medicine, Hazardous Substances and Health Science of the BGW and co-author of this publication. All other authors declare to have no potential conflict of interest. This does not alter our adherence to PLOS ONE policies on sharing data and materials.

## Introduction

Our society has to face the challenge of a growing number of older people [1], combined with an expected shortage of skilled workers, especially in nursing care [2]. At the same time, cancer patients, patients with non-oncological diseases, multimorbid patients [3] and patients suffering from dementia [4] are to benefit from palliative care. In Germany, palliative care is divided into specialised and general palliative care (Table 1). The German Society for Palliative Medicine (DGP) estimated that 90% of dying people are in need of palliative care, but only 10% of them are in need of specialised palliative care, because of more complex needs, such as complex pain management [5]. The framework of specialised palliative care encompasses specialist outpatient palliative care, inpatient hospices and palliative care units in hospitals. In Germany, most nurses in specialised palliative care have an additional qualification [6]. Further, nurses in specialist palliative care in Germany have fewer patients to care for than nurses in other fields which results in more time for the patients [7]. Most people are treated within general palliative care in non-specialized palliative care settings, which is provided by primary care suppliers with fundamental knowledge of palliative care. These are GPs, specialists (e.g. oncologists) and, above all, staff in nursing homes, hospitals and outpatient care [8]. Nurses in general palliative care have basic skills in palliative care from their education. However, there is no data available on the extent of palliative care they provide, or information on an additional qualification in palliative care. Palliative care experts from around the world consider the education and training of all staff in the fundamentals of palliative care to be essential [9] and a study conducted in Italy revealed that professional competency of palliative care nurses was positively associated with job satisfaction [10]. Thus, it is possible that the extent of palliative care or an additional qualification in palliative care may have implications on the working situation and health status of nurses. In Germany, there are different studies which concentrate on people dying in hospitals or nursing homes and the associated burden on the institution's staff [11, 12], but studies considering palliative care aspects concentrate on specialised palliative care settings [6, 13, 14]. Because the working conditions of nurses in specialised and general palliative care are somewhat different, as stated above, this paper focuses on nurses working in general palliative care, in other words, in non-specialised palliative care settings.

Burnout is a large problem in social professions, especially in health care worldwide [19] and is consistently associated with nurses intention to leave their profession [20]. Burnout is a state of emotional, physical, and mental exhaustion caused by a long-term mismatch of the demands associated with the job and the resources of the worker [21]. One of the causes for the alarming increase in nursing burnout is their workload [22, 23]. Workload can be either qualitative (pertaining to the type of skills and/or effort needed in order to perform work tasks) or quantitative (the amount of work to be done and the speed at which it has to be performed) [24].

**Table 1. General and specialised palliative care in Germany.**

|  | General palliative care | Specialised palliative care |
|---|---|---|
| **Outpatient care** | ▪ Outpatient care | ▪ Specialist outpatient palliative care |
| **Inpatient care** | ▪ Hospitals<br>▪ Nursing homes | ▪ Palliative care units<br>▪ Inpatient hospices |

*Note.* General palliative care in Germany also includes ambulatory hospice services (main characteristic is performance of volunteer work), palliative hospital beds, not specialised palliative care units or palliative medicine services in hospitals. These services of general palliative care were not included in this study. Specialised palliative care also includes specialised outpatient facilities, specialised palliative medicine services in hospitals and palliative day care clinics [15–18].

Studies analysing burnout in nursing have recognised different coping strategies, self-efficacy, emotional intelligence factors, social support [25, 26], the meaning of work and role clarity [27] as protective factors. Studies conducted in the palliative care sector identified empathy [28], attitudes toward death, secure attachment styles, and meaning and purpose in life as protective factors [29]. Individual factors such as spirituality and hobbies [30], self-care [31], coping strategies for facing the death of a patient [32], physical activity [33] and social resources, like social support [33, 34], the team [6, 13] and time for patients [32] were identified, as effectively protecting against burnout. These studies used qualitative or descriptive methods or correlation analyses in order to investigate the relationship between variables. In contrast to this statistical approach, fewer studies examined the buffering/moderating role of resources on the relationship between workload and burnout in nursing. A moderator variable affects the direction and/or the strength of the relationship between two other variables [35]. A previous study has showed resilience as being a moderator for emotional exhaustion on health [36], and other studies revealed professional commitment or social support moderating job demands on emotional exhaustion [37, 38]. Furthermore, work engagement and emotional intelligence was recognised as a moderator in the work demand and burnout relationship [39, 40].

We have analysed the working situation of nurses using the Rudow Stress-Strain-Resources model [41]. According to this model, the same stressor can lead to different strains in different people depending on available resources. These resources can be either individual, social or organisational. Individual resources are those resources which are owned by an individual. This includes for example personal capacities such as positive thinking as well as personal qualifications. Social resources consist of the relationships an individual has, this includes for example relationships at work as well as in his private life. Organisational resources refer to the concrete design of the workplace and work organisation. For example, nurses reporting a good working team may experience workload as less threatening and disruptive because a good working team gives them a feeling of security, stability and belonging. According to Rudow, individual, social or organisational resources can buffer/moderate the negative effects of job demands (stressors) on, for example, burnout (strain).

Nurses' health may have an effect on the quality of the services offered by the health care system [42], therefore, it is of great interest to do everything possible to preserve their health. This may be achieved by reducing the workload and by strengthening the available resources. However, to the best of our knowledge, we are not aware of any study which considers palliative care aspects within general palliative care in Germany. Therefore, the aim of the study was to investigate the buffering role of resources on the relationship between workload ('quantitative demands') and burnout among nurses. Palliative care aspects, such as information on the extent of palliative care were incorporated to the analyses as covariates.

## Methods

### Study design and participants

An exploratory cross-sectional study was conducted in 2017. In Germany, there is no national register for nurses. Data for this study were collected from a stratified 10% random sample of a database with outpatient facilities, hospitals and nursing homes in Germany from the Institution for Statutory Accident Insurance and Prevention in Health and Welfare Services in Germany. This institution is part of the German social security system. It is the statutory accident insurer for nonstate institutions in the health and welfare services in Germany and thus responsible for the health concerns of the target group investigated in the present study, namely nurses. Due to data protection rules, this institution was also responsible for the first contact with the health facilities. 126 of 3,278 (3.8%) health facilities agreed to participate in

the survey. They informed the study team about how many nurses worked in their institution, and whether the nurses would prefer to answer a paper-and-pencil questionnaire (with a pre-franked envelope) or an online survey (with an access code). 2,982 questionnaires/access codes were sent out to the participating health facilities (656 to outpatient care, 160 to hospitals and 2,166 to nursing homes), where they were distributed to the nurses (S1 Table). Participation was voluntary and anonymous. Informed consent was obtained written at the beginning of the questionnaire. Approval to perform the study was obtained by the ethics committee of the State Chamber of Medicine in Rhineland-Palatinate (Clearance number 837.326.16 (10645)).

## Questionnaire

The questionnaire contained questions regarding i) nurse's sociodemographic information and information on current profession as well as ii) palliative care aspects. Furthermore, iii) parts of the German version of the Copenhagen Psychosocial Questionnaire (COPSOQ), iv) a resilience questionnaire [RS-13] and v) single questions relating to resources were added.

**i) Sociodemographic information and information on current profession.** The nurse's sociodemographic information and information on current profession included the variables 'age', 'gender', 'marital status', 'education', 'professional qualification', 'working area', 'professional experience' and 'extent of employment'.

**ii) Palliative care aspects.** Palliative care aspects included self-developed questions on 'additional qualification in palliative care', the 'number of patients' deaths within the last month (that the nurses cared for personally)' and the 'extent of palliative care'. The latter was evaluated by asking: how much of your working time (as a percentage) do you spend with care of palliative patients? The first two items were already used in the pilot study. The pilot study consisted of a qualitative part, where interviews with experts in general and specialised palliative care were performed [43]. These interviews were used to develop a standardized questionnaire which was used for a cross-sectional pilot survey [6, 44].

**iii) Copenhagen Psychosocial Questionnaire (COPSOQ).** The questionnaire included parts of the German standard version of the Copenhagen Psychosocial Questionnaire (COPSOQ) [45]. The COPSOQ is a valid and reliable questionnaire for the assessment of psychosocial work environmental factors and health in the workplace [46, 47]. The scales selected were 'quantitative demands' (four items, for example: "Do you have to work very fast?") measuring workload, 'burnout' (six items, for example: "How often do you feel emotionally exhausted?"), 'meaning of work' (three items, for example: "Do you feel that the work you do is important?") and 'workplace commitment' (four items, for example: "Do you enjoy telling others about your place of work?").

**iv) Resilience questionnaire RS-13.** The RS-13 questionnaire is the short German version of the RS-25 questionnaire developed by Wagnild & Young [48]. The questionnaire postulates a two-dimensional structure of resilience formed by the factors "personal competence" and "acceptance of self and life". The RS-13 questionnaire measures resilience with 13 items on a 7-point scale (1 = I do not agree, 7 = I totally agree with different statements) and has been validated in representative samples [49, 50]. The results of the questionnaire were grouped into persons with low, moderate or high resilience.

**v) Questions on resources.** Single questions on personal, social and organizational resources assessed the nurses' views of these resources in being helpful in dealing with the demands of their work. Further, single questions collected the agreement to different statements such as 'Do you receive recognition for your work from the supervisor?' (see Table 4). These resources were frequently reported in the pilot study by nurses in specialised palliative care [6].

## Data preparation and analysis

The data from the paper-and-pencil and online questionnaires were merged, and data cleaning was done (e.g. questionnaires without specification to nursing homes, hospitals or outpatient care were excluded). The scales selected from the COPSOQ were prepared according to the COPSOQ guidelines. In general, COPSOQ items have a 5-point Likert format, which are then transformed into a 0 to 100 scale. The scale score is calculated as the mean of the items for each scale, if at least half of the single items had valid answers. Nurses who answered less than half of the items in a scale were recorded as missing. If at least half of the items were answered, the scale value was calculated as the average of the items answered [46]. High values for the scales 'quantitative demands' and 'burnout' were considered negative, while high values for the scales 'meaning of work' and 'workplace commitment' were considered positive. The proportion of missing values for single scale items was between 0.5% and 2.7%. Cronbach's Alpha was used to assess the internal consistency of the scales. A Cronbach's Alpha > 0.7 was regarded as acceptable [35]. The score of the RS-13 questionnaire ranges from 13 to 91. The answers were grouped according to the specifications in groups with low resilience (score 13–66), moderate resilience (67–72) and high resilience (73–91) [49]. The categorical resource variables were dichotomised (example: not helpful/little helpful vs. quite helpful/very helpful).

The study was conceptualised as an exploratory study. Consequently, no prior hypotheses were formulated, so the p-values merely enable the recognition of any statistically noteworthy findings [51]. Descriptive statistics (absolute and relative frequency, M = mean, SD = standard deviation) were used to depict the data. Bivariate analyses (Pearson correlation, t-tests, analysis of variance) were performed to infer important variables for the regression-based moderation analysis. Variables which did not fulfil all the conditions for linear regression analysis were recoded as categorical variables [35]. The variable 'extent of palliative care' was categorised as '≤ 20 percent of working time' vs. '> 20 percent of working time' due to the median of the variable (median = 20).

The first step with regard to the moderation analysis was to determine the resource variables. Therefore all resource variables that reached a p-value < 0.05 in the bivariate analysis with the scale 'burnout' were further analysed (scale 'meaning of work', scale 'workplace commitment', variables presented in Table 4). The moderator analysis was conducted using the PROCESS program developed by Andrew F. Hayes. First, scales were mean-centred to reduce possible scaling problems and multicollinearity. Secondly, for all significant resource variables the following analysis were done: the 'quantitative demand', one resource (one per model) and the interaction term between the 'quantitative demand' and the resource, as well as the covariates 'age', 'gender', 'working area', 'extent of employment', the 'extent of palliative care' and the 'number of patient deaths within the last month' were added to the moderator analysis, in order to control for confounding influence. If the interaction term between the 'quantitative demand' and the resource accounted for significantly more variance than without interaction term (change in $R^2$ denoted as $\Delta R^2$, $p < 0.05$), a moderator effect of the resource was present. The interaction of the variables (± 1 SD the mean or variable manifestation such as yes and no) was plotted.

All the statistical calculations were performed using the Statistical Package for Social Science (SPSS, version 23.5) and the PROCESS macro for SPSS (version 3.5 by Hayes) for the moderator analysis.

## Results

Of the 2,982 questionnaires/access codes sent out, 497 were eligible for the analysis. The response rate was 16.7% (response rate of outpatient care 14.6%, response rate of hospitals

18.1% and response rate of nursing homes 16.0%). Since only n = 29 nurses from hospitals participated, these were excluded from data analysis. After data cleaning, the final number of participants was n = 437.

## Descriptive results

The basic characteristics of the study population are presented in Table 2. The average age of the nurses was 42.8 years, and 388 (89.6%) were female. In total, 316 nurses answered the question how much working time they spend caring for palliative patients. Sixteen (5.1%) nurses reported spending no time caring for palliative patients, 124 (39.2%) nurses reported between 1% to 10%, 61 (19.30%) nurses reported between 11% to 20% and 115 (36.4%) nurses reported spending more than 20% of their working time for caring for palliative patients. Approximately one-third (n = 121, 27.7%) of the nurses in this study did not answer this question. One hundred seventeen (29.5%) nurses reported 4 or more patient deaths, 218 (54.9%) reported 1 to 3 patient deaths and 62 (15.6%) reported 0 patient deaths within the last month.

Table 3 presents the mean values and standard deviations of the scales 'quantitative demands', 'burnout', and the resource scales 'meaning of work' and 'workplace commitment'. All scales achieved a satisfactory level of internal consistency.

## Bivariate analyses

There was a strong positive correlation between the 'quantitative demands' and 'burnout' scales (r = 0.498, p ≤ 0.01), and a small negative correlation between 'burnout' and 'meaning of work' (r = -0.222, p ≤ 0.01) and 'workplace commitment' (r = -0.240, p ≤ 0.01). Regarding the basic and job-related characteristics of the sample shown in Table 2, 'burnout' was significantly related to 'extent of palliative care' (≤ 20% of working time: n = 199, M = 46.06, SD = 20.28; > 20% of working time: n = 115, M = 53.80, SD = 20.24, t(312) = -3.261, p = 0.001). Furthermore, there was a significant effect regarding the 'number of patient deaths during the last month' (F (2, 393) = 5.197, p = 0.006). The mean of the burnout score was lower for nurses reporting no patient deaths within the last month than for nurses reporting four or more deaths (n = 62, M = 42.47, SD = 21.66 versus n = 116, M = 52.71, SD = 20.03). There was no association between 'quantitative demands' and an 'additional qualification in palliative care' (no qualification: n = 328, M = 55.77, SD = 21.10; additional qualification: n = 103, M = 54.39, SD = 20.44, p = 0.559).

The association between 'burnout' and the evaluated (categorical) resource variables is presented in Table 4. Nurses mostly had a lower value on the 'burnout' scale when reporting various resources. Only the resources 'family', 'religiosity/spirituality', 'gratitude of patients', 'recognition through patients/relatives' and an 'additional qualification in palliative care' were not associated with 'burnout'.

## Moderator analyses

In total, 16 moderation analyses were conducted. Table 5 presents the results of the moderation analyses where a significant moderation was found. For 'workplace commitment', there was a positive and significant association between 'quantitative demands' and 'burnout' (b = 0.47, SE = 0.051, p < 0.001). An increase of one value on the scale 'quantitative demands' increased the scale 'burnout' by 0.47. 'Workplace commitment' was negatively related to 'burnout', meaning that a higher degree of 'workplace commitment' was related to a lower level of 'burnout' (b = -0.11, SE = 0.048, p = 0.030). A model with the interaction term of 'quantitative demands' and the resource 'workplace commitment' accounted for significantly more variance in 'burnout' than a model without interaction term ($\Delta R^2$ = 0.021, p = 0.004).

**Table 2. Basic and job-related characteristics of the sample (n = 437).**

| Variable | |
|---|---|
| Age in years, mean (SD) | 42.8 (11.8) |
| Age grouped, no. (%) | |
| < 35 | 118 (27.7) |
| 35–49 | 154 (36.2) |
| ≥ 50 | 154 (36.2) |
| Gender, no. (%) | |
| male | 45 (10.4) |
| female | 388 (89.6) |
| Marital status, no. (%) | |
| single | 140 (32.6) |
| married | 210 (49.0) |
| divorces/widowed | 79 (18.4) |
| Education, no. (%) | |
| without a school-leaving qualification/ secondary school leaving certificate/ other qualification | 69 (16.0) |
| intermediate school-leaving certificate | 239 (55.3) |
| qualification for university entrance | 124 (28.7) |
| Professional qualification, no. (%) | |
| nursing assistant | 79 (18.6) |
| nurse | 75 (17.7) |
| geriatric nurse | 196 (46.2) |
| others (in training, other education) | 74 (17.5) |
| Working area, no. (%) | |
| nursing home | 344 (78.7) |
| outpatient care | 93 (21.3) |
| Professional experience in years, mean (SD) | 14 (10.6) |
| Extent of employment, no. (%) | |
| part-time job | 175 (40.4) |
| full-time job | 258 (59.6) |
| Additional qualification in palliative care, no. (%) | |
| no | 329 (76.2) |
| yes/ currently absolving furhter qualification | 103 (23.8) |
| Extent of palliative care (as percentage), no. (%) | |
| ≤ 20 of working time | 201 (63.6) |
| > 20 of working time | 115 (36.4) |
| Number of patient deaths (in the last month), no. (%) | |
| 0 | 62 (15.6) |
| 1–3 | 218 (54.9) |
| ≥ 4 | 117 (29.5) |

*Note*. Shown are valid percentages; Missing values: age (n = 11), sex (n = 4), marital status (n = 8), education (n = 5), professional qualification (n = 13), professional experience (n = 16), extent of employment (n = 4), additional qualification in palliative care (n = 5), extent of palliative care (n = 121), number of patient deaths (n = 40)

The impact of 'quantitative demands' on 'burnout' was dependent on 'workplace commitment' (b = -0.01, SE = 0.002 p = 0.004). The variables explained 31.9% of the variance in 'burnout'.

Regarding the 'good working team' resource, the variables 'quantitative demands' and 'burnout' were positively and significantly associated (b = 0.76, SE = 0.154, p < 0.001), and the

**Table 3. Means and standard deviations of COPSOQ scales.**

| Variable | Number of items | Cronbach's Alpha | n | M (SD) | Range |
|---|---|---|---|---|---|
| Quantitative demands | 4 | 0.798 | 436 | 55.38 (20.83) | 0–100 |
| Burnout | 6 | 0.907 | 435 | 48.77 (20.28) | 0–100 |
| Meaning of work | 3 | 0.827 | 430 | 82.17 (18.76) | 0–100 |
| Workplace commitment | 4 | 0.711 | 430 | 56.27 (22.03) | 0–100 |

*Note*. M = mean, SD = standard deviation

variables 'good working team' and 'burnout' were not associated (b = -3.15, SE = 3.52, p = 0.372). A model with the interaction term of 'quantitative demands' and the 'good working team' resource accounted for significantly more variance in 'burnout' than a model without interaction term ($\Delta R^2$ = 0.011, p = 0.040). The 'good working team' resource moderated the impact of 'quantitative demands' on 'burnout' (b = -0.34, SE = 0.165, p = 0.004). The variables explained 29.7% of the variance in 'burnout'.

The associations between 'quantitative demands' and 'burnout' (b = 0.63, SE = 0.085, p < 0.001), between 'recognition supervisor' and 'burnout' (b = -7.29, SE = 2.27, p = 0.001), and the interaction term of 'quantitative demands' and the resource 'recognition supervisor' (b = -0.34, SE = 0.108, p = 0.002) were significant. Again, a model with the interaction term accounted for significantly more variance in 'burnout' than a model without interaction term ($\Delta R^2$ = 0.024, p = 0.002). 'Recognition from supervisor' influenced the impact of 'quantitative demands' on burnout for -0.34 on the 0 to 100 scale. The variables explained 33.7% of the variance in 'burnout'.

Figs 1–3 demonstrates simple slopes of the interaction effects of 'workplace commitment' predicting 'burnout' at high, average and low levels (Fig 1) respectively with and without the resource 'good working team' (Fig 2) and 'recognition from supervisor' (Fig 3). Higher 'quantitative demands' were associated with higher levels of 'burnout'. At low 'quantitative demands', the 'burnout' level was quite similar for all nurses. However, when 'quantitative demands' increased, nurses who confirmed that they had the resources stated a lower 'burnout' level than nurses who denied having them. This trend is repeated by the resources 'workplace commitment', 'good working team' and 'recognition from supervisor'.

The palliative care aspect 'extent of palliative care' showed that spending more than 20 percent of working time in care for palliative patients increased burnout significantly by a value of approximately 5 on a 0 to 100 scale (Table 5).

## Discussion

The aim of the present study was to analyse the buffering role of resources on the relationship between workload and burnout among nurses. This was done for the first time by considering palliative care aspects, such as information on the extent of palliative care.

The study shows that higher quantitative demands were associated with higher levels of burnout, which is in line with other studies [37, 39]. Furthermore, the results of this study indicate that working in a good team, recognition from supervisor and workplace commitment is a moderator within the workload—burnout relationship. Although the moderator analyses revealed low buffering effect values, social resources were identified once more as important resources. This is consistent with the results of a study conducted in the field of specialised palliative care in Germany, where a good working team and workplace commitment moderated the impact of quantitative demands on nurses burnout [52]. A recently published review also

**Table 4. Results of bivariate analysis of resources (categorical variables) and burnout.**

| Variables | | n | M (SD) | t | df | p |
|---|---|---|---|---|---|---|
| **Personal/social resources** | | | | | | |
| Family | not/little helpful | 54 | 54.63 (23.41) | 1.903 | 64.3 | 0.062 |
| | quite/very helpful | 371 | 48.26 (19.66) | | | |
| Friends | not/little helpful | 66 | 56.63 (20.19) | 3.346 | 423 | 0.001** |
| | quite/very helpful | 359 | 47.64 (20.04) | | | |
| Positive thinking | not/little helpful | 76 | 60.92 (19.02) | 5.860 | 423 | < 0.001** |
| | quite/very helpful | 349 | 46.41 (19.68) | | | |
| Professional attitude/ dissociation | not/little helpful | 79 | 54.64 (21.40) | 2.726 | 423 | 0.007** |
| | quite/very helpful | 346 | 47.81 (19.78) | | | |
| Hobbies | not/little helpful | 95 | 55.93 (21.60) | 3.852 | 419 | < 0.001** |
| | quite/very helpful | 326 | 46.96 (19.48) | | | |
| Self-care | not/little helpful | 100 | 56.79 (20.36) | 4.494 | 419 | < 0.001** |
| | quite/very helpful | 321 | 46.57 (19.72) | | | |
| Self-reflection | not/little helpful | 116 | 53.32 (20.69) | 2.780 | 417 | 0.006** |
| | quite/very helpful | 303 | 47.22 (19.85) | | | |
| Sport | not/little helpful | 234 | 50.75 (20.36) | 2.106 | 419 | 0.036* |
| | quite/very helpful | 187 | 46.57 (20.06) | | | |
| Religiosity/ spirituality | not/little helpful | 314 | 48.72 (20.26) | -0.564 | 420 | 0.573 |
| | quite/very helpful | 108 | 50.00 (20.72) | | | |
| Resilience | low/ moderate | 181 | 55.57 (18.64) | 6.072 | 403 | < 0.001** |
| | high | 224 | 43.68 (20.32) | | | |
| **Organisational resources** | | | | | | |
| Working in a good team | do not agree/rather disagree | 55 | 60.00 (21.69) | 4.478 | 428 | < 0.001** |
| | somewhat agree/fully agree | 375 | 47.19 (19.52) | | | |
| Gratitude of patients | do not agree/rather disagree | 41 | 54.67 (23.72) | 1.688 | 46.0 | 0.098 |
| | somewhat agree/fully agree | 390 | 48.20 (19.75) | | | |
| Gratitude of relatives | do not agree/rather disagree | 63 | 56.75 (22.57) | 3.078 | 78.7 | 0.003** |
| | somewhat agree/fully agree | 367 | 47.45 (19.53) | | | |
| Recognition from patients/ relatives | do not agree/rather disagree | 66 | 52.84 (22.28) | 1.739 | 426 | 0.083 |
| | somewhat agree/fully agree | 362 | 48.14 (19.80) | | | |
| Recognition from colleagues | do not agree/rather disagree | 77 | 59.81 (21.23) | 5.376 | 427 | < 0.001** |
| | somewhat agree/fully agree | 352 | 46.53 (19.26) | | | |
| Recognition through social context | do not agree/rather disagree | 70 | 53.81 (21.02) | 2.270 | 424 | 0.024* |
| | somewhat agree/fully agree | 356 | 47.83 (19.99) | | | |
| Recognition from supervisor | do not agree/rather disagree | 162 | 57.54 (19.44) | 7.265 | 423 | < 0.001** |
| | somewhat agree/fully agree | 263 | 43.63 (19.00) | | | |
| Recognition through salary | do not agree/rather disagree | 300 | 51.51 (19.80) | 4.211 | 426 | < 0.001** |
| | somewhat agree/fully agree | 128 | 42.68 (20.02) | | | |
| Additional qualification in palliative care | no | 328 | 48.96 (20.39) | 0.520 | 428 | 0.603 |
| | yes/ currently absolving furhter qualification | 102 | 47.76 (20.20) | | | |

*Note.* M = mean, SD = standard deviation,

*p ≤ 0.05,

**p ≤ 0.01

**Table 5. Coefficients of the moderated regression model for burnout.**

| | | Workplace commitment | | | | Good working team | | | | Recognition from supervisor | | | |
|---|---|---|---|---|---|---|---|---|---|---|---|---|---|
| | | b | se | t | p | b | se | t | p | b | se | t | p |
| Age | < 35 | -2.11 [-7.23, 3.01] | 2.601 | -0.81 | 0.418 | -2.38 [-7.53, 2.78] | 2.618 | -0.91 | 0.365 | -2.75 [-7.82, 2.33] | 2.579 | -1.06 | 0.288 |
| | 35–49 | -1.23 [-5.99, 3.54] | 2.420 | -0.51 | 0.612 | -0.73 [-5.57, 4.11] | 2.461 | -0.30 | 0.766 | -1.17 [-5.92, 3.58] | 2.41 | -0.48 | 0.629 |
| | $\geq 50$ | Ref. | | | | Ref. | | | | Ref. | | | |
| Sex | male | Ref. | | | | Ref. | | | | Ref. | | | |
| | female | 2.98 [-3.61, 9.56] | 3.344 | 0.89 | 0.374 | 4.00 [-2.82, 10.82] | 3.466 | 1.15 | 0.249 | 2.64 [-4.03, 9.31] | 3.388 | 0.78 | 0.437 |
| Working area | nursing home | Ref. | | | | Ref. | | | | Ref. | | | |
| | outpatient care | 0.27 [-2.34, 2.88] | 1.327 | 0.20 | 0.839 | -0.62 [-3.25, 2.02] | 1.34 | -0.46 | 0.647 | -0.37 [-2.98, 2.25] | 1.329 | -0.27 | 0.784 |
| Extent of employment | part-time job | Ref. | | | | Ref. | | | | Ref. | | | |
| | full-time job | -0.35 [-4.69, 3.99] | 2.204 | -0.16 | 0.873 | -0.06 [-4.48, 4.35] | 2.242 | 0.03 | 0.978 | -0.98 [-5.28, 3.33] | 2.185 | -0.45 | 0.655 |
| Extent of palliative care | $\leq 20$ | Ref. | | | | Ref. | | | | Ref. | | | |
| | > 20 | 4.68 [0.38, 8.98] | 2.183 | 2.14 | 0.033* | 5.04 [0.70, 9.38] | 2.205 | 2.28 | 0.023* | 4.39 [0.12, 8.66] | 2.171 | 2.02 | 0.044* |
| Number of patients deaths during the last month | 0 | Ref. | | | | Ref. | | | | Ref. | | | |
| | 1–3 | -0.06 [-5.67, 5.78] | 2.906 | 0.02 | 0.985 | 1.40 [-4.35, 7.15] | 2.920 | 0.48 | 0.633 | 1.64 [-4.01, 7.29] | 2.870 | 0.57 | 0.568 |
| | $\geq 4$ | 0.89 [-5.50, 7.29] | 3.248 | 0.28 | 0.784 | 2.20 [-4.35, 8.75] | 3.328 | 0.66 | 0.509 | 2.13 [-4.22, 8.48] | 3.225 | 0.66 | 0.510 |
| Quantitative demands (QD) | | 0.47 [0.37, 0.57] | 0.051 | 9.20 | <0.001** | 0.76 [0.45, 1.06] | 0.154 | 4.924 | <0.001** | 0.63 [0.46, 0.80] | 0.085 | 7.37 | <0.001** |
| Workplace commitment (WC) | | -0.11 [-0.20, -0.01] | 0.048 | -2.18 | 0.030* | | | | | | | | |
| Interaction QD * WC | | -0.01 [-0.01, 0.002] | 0.002 | -2.92 | 0.004** | | | | | | | | |
| Good working team (GWT) | not agree/ rather disagree | | | | | Ref. | | | | | | | |
| | agree/ fully agree | | | | | -3.15 [-10.07, 3.79] | 3.520 | -0.89 | 0.372 | | | | |
| Interaction QD * GWT | | | | | | -0.34 [-0.67, -0.02] | 0.165 | -2.063 | <0.040* | | | | |
| Recognition from supervisor (RFS) | not agree/ rather disagree | | | | | | | | | Ref. | | | |
| | agree/ fully agree | | | | | | | | | -7.29 [-11.67, -2.91] | 2.227 | -3.27 | 0.001** |
| Interaction QD * RFS | | | | | | | | | | -0.34 [-0.55, -0.13] | 0.108 | -3.17 | 0.002** |

*Note*. Workplace commitment: $R^2 = 0.319$, $F(11, 279) = 11.856$, $p < 0.001$, good working team: $R^2 = 0.297$, $F(11, 280) = 10.779$, $p < 0.001$, recognition through supervisor: $R^2 = 0.337$, $F(11, 276) = 12.742$, $p < 0.001$,

*$p \leq 0.05$,

**$p \leq 0.01$

describes social support from co-workers and supervisors as a fundamental resource in preventing burnout in nurses [53]. Workplace commitment was not only reported as a moderator between workload and health in the nurse setting [37], but also as a moderator between work stress and burnout [54] and between work stress and other health related aspects outside the

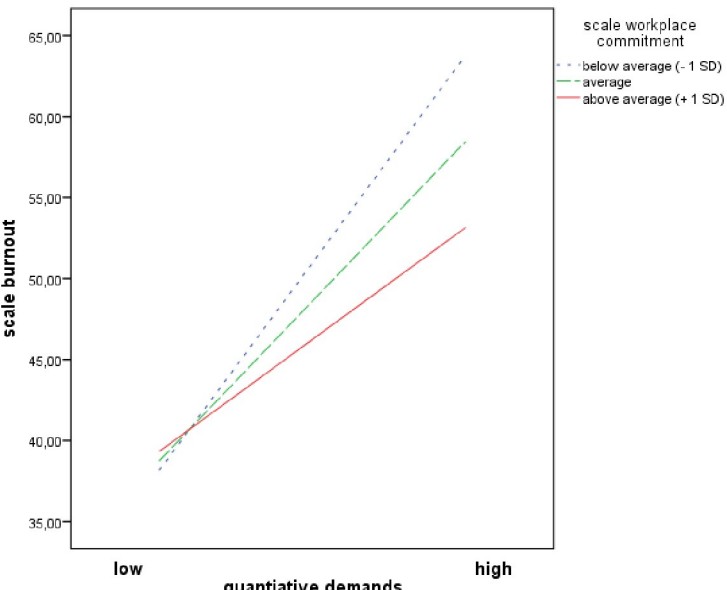

**Fig 1. Moderator effects of 'workplace commitment' on quantitative demands and burnout relationship.**

nurse setting [55]. In the present study, the effect of high workload on burnout was reduced with increasing workplace commitment. Nurses reporting a high work commitment may experience workload as less threatening and disruptive because workplace commitment gives them a feeling of belonging, security and stability. However, there are also some correlation studies which observed no direct relationship between workplace commitment and burnout for occupations in the health sector [56]. A study from Serbia assessed workplace commitment by nurses and medical technicians as a protective factor against patient-related burnout, but not against personal and work-related burnout [57]. Furthermore, a study conducted in

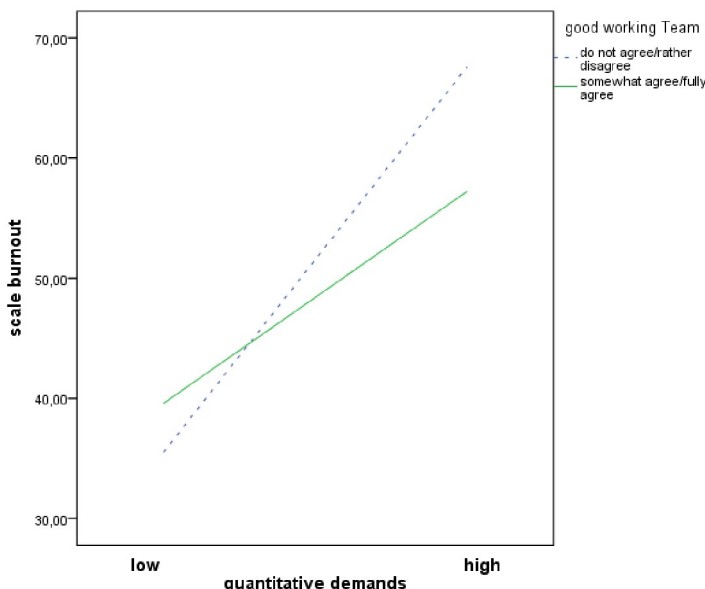

**Fig 2. Moderator effects of 'good working team' on quantitative demands and burnout relationship.**

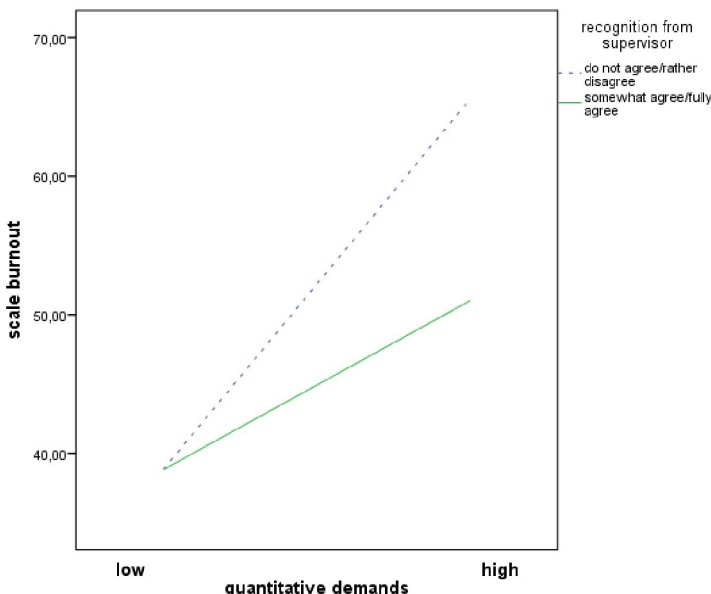

**Fig 3. Moderator effects of 'recognition from supervisor' on quantitative demands and burnout relationship.**

Estonia reported no relationship between workplace commitment and burnout amongst nurses [58]. As there are indications that workplace commitment is correlated with patient safety [59], the development and improving of workplace commitment needs further scientific investigation.

This study observed slightly higher burnout rates among nurses who reported a 'good working team' for low workload. This fact is not decisive for the interpretation of the moderation effect of this resource because moderation is present. When workload increased, nurses who confirmed that they worked in a good working team stated a lower burnout level. However, the result of the current study showed that a good working team is particularly important when workload increases, in the most extreme cases team work in palliative care is necessary to save a person's life. Because team work in today's health care system is essential, health care organisations should foster team work in order to enhance their clinical outcomes [60], improve the quality of patient care as well as health [61] and satisfaction of nurses [62].

The bivariate analysis revealed that nurses who reported getting recognition from colleagues, through the social context, salary and gratitude from relatives of patients stated a lower value on the burnout scale. This is in accordance with the results of a qualitative study, which indicated that the feeling of recognition, and that one's work is useful and worthwhile, is very important for nurses and a source of satisfaction [63]. Furthermore, self-care, self-reflection [64] and professional attitude/dissociation seem to play an important role in preventing burnout. The bivariate analysis also revealed a relationship between resilience and burnout. Nurses with high resilience reported lower values on the burnout scale, but a buffering role of resilience on burnout was not assessed. The present paper focuses solely on quantitative demands and burnout. In future studies, the different fields of nursing demands, like organisational or emotional demands, should be assessed in relation to burnout, job satisfaction and health.

Finally, we observed whether the consideration of palliative care aspects is associated with burnout. The bivariate analysis revealed a relationship between the extent of palliative care, number of patient deaths within the last month and burnout. Using regression analyses, only

the extent of palliative care was associated with burnout. Since, to the best of our knowledge, the present study is the first study to consider palliative care aspects within general palliative care in Germany, these variables need further scientific investigation, not only within the demand—burnout relationship but also between the demand—health and the demand—job satisfaction relationship. Furthermore, palliative care experts from around the world considered the education and training of all members of staff in the fundamentals of palliative care to be essential [9]. One-fourth of the respondents in the present study had an additional qualification in palliative care, which was not obligatory. We assessed a relationship between quantitative demands and burnout but no relationship between an additional qualification and quantitative demands nor burnout. Nevertheless, we assessed a protective effect of the additional qualification within the pilot study in specialised palliative care, in relation both to organisational demands and demands regarding the care of relatives [6]. This suggests that the additional qualification is a resource, but one which depends on the field of demand. Further analyses would be required to review benefits achieved by additional qualifications in general palliative care.

The variable extent of palliative care is the one with the most missing values in the survey, thus future analyses should not only study larger samples but also reconsider the question on extent of palliative care.

Finally, it can be said that the main contribution of the present study is to make palliative care aspects in non-specialised palliative care settings a subject of discussion.

## Limitations

The following potential limitations need to be stated: although a random sample was drawn, the sample is not representative for general palliative care in Germany due to a low participation rate of the health facilities, a low response rate of the nurses, the different responses of the health facilities and the exclusion of hospitals. One possible explanation for the low participation rate of the health facilities is the sampling procedure and data protection rules, which did not allowed the study team to contact the institutions in the sample. Due to the low participation rate, the results of the present study may be labelled as preliminary. Further, the data are based on a detailed and anonymous survey, and therefore the potential for selection bias has to be considered. It is possible that the institutions and nurses with the highest burden had no time for or interest in answering the questionnaire. It is also possible that the institutions which care for a high number of palliative patients may have taken particular interest in the survey. Additionally, some items of the questionnaire were self-developed and not validated but were considered valuable for our study as they answered certain questions that standardized questionnaires could not. The moderator analyses revealed low effect values and the variance explained by the interaction terms is rather low. However, moderator effects are difficult to detect, therefore, even those explaining as little as one percent of the total variance should be considered [65]. Consequently, the additional amount of variance explained by the interaction in the current study (2% for workplace commitment and recognition of supervisor and 1% for good working team) is not only statistically significant but also practically and theoretically relevant. When considering the results of the current study, it must be taken into account that the present paper focuses solely on quantitative demands and burnout. In future studies, the different fields of nursing demands have to be carried out on the role of resources. This not only pertains for burnout, but also for other outcomes such as job satisfaction and health. Finally, the cross-sectional design does not allow for casual inferences. Longitudinal and interventional studies are needed to support causality in the relationships examined.

## Conclusions

The present study provides support to a buffering role of workplace commitment, good working teams and recognition from supervisors on the relationship between workload and burnout. Initiatives to develop or improve workplace commitment and strengthen collaboration with colleagues and supervisors should be implemented in order to reduce burnout levels. Furthermore, the results of the study provides first insights that palliative care aspects in general palliative care may have an impact on nurse burnout, and therefore they have gone unrecognised for too long in the scientific literature. They have to be considered in future studies, in order to improve the working conditions, health and satisfaction of nurses. As our study was exploratory, the results should be confirmed in future studies.

## Supporting information

**S1 Table. Number of questionnaires sent out to facilites and response rate.**
(DOCX)

## Acknowledgments

We thank the nurses and the health care institutions for taking part in the study. We thank D. Wendeler, O. Kleinmüller, E. Muth, R. Amma and C. Kohring who were helpful in the recruitment of the participants and data collection.

## Author Contributions

**Conceptualization:** Elisabeth Diehl, Sandra Rieger, Stephan Letzel, Anja Schablon, Albert Nienhaus, Luis Carlos Escobar Pinzon.

**Data curation:** Elisabeth Diehl, Sandra Rieger.

**Formal analysis:** Elisabeth Diehl.

**Funding acquisition:** Stephan Letzel, Luis Carlos Escobar Pinzon.

**Investigation:** Elisabeth Diehl, Sandra Rieger, Stephan Letzel, Anja Schablon, Albert Nienhaus, Luis Carlos Escobar Pinzon.

**Methodology:** Elisabeth Diehl, Sandra Rieger, Anja Schablon, Albert Nienhaus, Luis Carlos Escobar Pinzon.

**Project administration:** Elisabeth Diehl, Sandra Rieger, Anja Schablon, Luis Carlos Escobar Pinzon.

**Resources:** Elisabeth Diehl, Sandra Rieger, Stephan Letzel, Anja Schablon, Albert Nienhaus, Luis Carlos Escobar Pinzon.

**Software:** Elisabeth Diehl, Sandra Rieger.

**Supervision:** Anja Schablon, Albert Nienhaus, Luis Carlos Escobar Pinzon, Pavel Dietz.

**Validation:** Elisabeth Diehl.

**Visualization:** Elisabeth Diehl.

**Writing – original draft:** Elisabeth Diehl.

**Writing – review & editing:** Elisabeth Diehl, Sandra Rieger, Stephan Letzel, Anja Schablon, Albert Nienhaus, Luis Carlos Escobar Pinzon, Pavel Dietz.

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
