## [Decision Letter · Decision Letter 0]

18 Sep 2020

PONE-D-20-23720

The relationship between workload and burnout among nurses in non-specialized palliative care settings: the buffering role of personal, social and organisational resources

PLOS ONE

Dear Dr. Dietz,

Thank you for submitting your manuscript to PLOS ONE. After careful consideration, we feel that it has merit but does not fully meet PLOS ONE’s publication criteria as it currently stands. Therefore, we invite you to submit a revised version of the manuscript that addresses the points raised during the review process.

We look forward to receiving your revised manuscript.

Kind regards,

Prof. Dr. Adrian Loerbroks

Academic Editor

PLOS ONE

'I have read the journal's policy and the authors of this manuscript have the following competing interests: The project was funded by the BGW - Berufsgenossenschaft für Gesundheitsdienst und Wohlfahrtspflege (Institution for Statutory Accident Insurance and Prevention in Health and Welfare Services). The BGW is responsible for the health concerns of the target group investigated in the present study, namely nurses. Prof. Dr. A. Nienhaus is head of the Department for Occupational Medicine, Hazardous Substances and Health Science of the BGW and co-author of this publication. All other authors declare to have no potential conflict of interest.  '

a. Please confirm that this does not alter your adherence to all PLOS ONE policies on sharing data and materials, by including the following statement: "This does not alter our adherence to  PLOS ONE policies on sharing data and materials.” (as detailed online in our guide for authors http://journals.plos.org/plosone/s/competing-interests).  If there are restrictions on sharing of data and/or materials, please state these.

Please note that we cannot proceed with consideration of your article until this information has been declared.

Reviewers' comments:

Reviewer's Responses to Questions

**Comments to the Author**

1. Is the manuscript technically sound, and do the data support the conclusions?

Reviewer #1: Yes

Reviewer #2: Partly

Reviewer #3: Partly

2. Has the statistical analysis been performed appropriately and rigorously? 

Reviewer #1: Yes

Reviewer #2: Yes

Reviewer #3: I Don't Know

3. Have the authors made all data underlying the findings in their manuscript fully available?

Reviewer #1: Yes

Reviewer #2: No

Reviewer #3: No

4. Is the manuscript presented in an intelligible fashion and written in standard English?

Reviewer #1: Yes

Reviewer #2: No

Reviewer #3: Yes

5. Review Comments to the Author

Reviewer #1: Congratulations to the authors for their work. It is a relevant topic and has an impact on clinical practice.

The limitations that I identified in the studies were assumed by the authors in the limitations section.

In the methodological part, although the tests used are perceived, it was not clear to me the assumptions behind them (parametric studies vs. nonparametric studies), how this option was assumed.

The discussion can be further developed to highlight the results obtained in the light of the existing evidence.

Reviewer #2: The present study deals with burnout in the context of nursing and the possible moderating role of several personal, social, and organizational resources on the relationship between quantitative demands and burnout. It is the first study to also include palliative care aspects such as the ‚extent of palliative care‘ within different nursing tasks and the ‚number of patient deaths during the last month‘. The results of the study indicate a moderating role of workplace commitment, a good working team and recognition of one’s supervisor on the relationship between quantitative demands and burnout among nurses.

A very comprehensive questionnaire is used including many validated scales such as parts of the Copenhagen Psychosocial Questionnaire and the RS-13. The study design seems appropriate in order to answer the research question.

Yet, some major limitations apply. All in all, the manuscript is somewhat confusing to read, which is not least due to many linguistic irregularities, but also to a partly imprecise and sketchy presentation of the methodology. Furthermore, the study sample is not representative for nurses working in palliative care in Germany and the response rate of 3.8% is exceptionally low.

Please find below my specific comments to the manuscript:

#1 I assume that the manuscript has not been reviewed by a native speaker. Both linguistic oddities and incorrect use of grammar and verbs are noticeable. Examples:

Line 66 possessive apostrophe missing (it should be nurses’ instead of nurses)

Line 68/69 use ‘alarming increase’ instead of ‘worrying development’

Line 71 s is missing (it should be concerns instead of concern)

Line 77/78 use ‘such as’ or ‘e.g.’ instead of ‘like’

Line 137 sudden change from past tense to present tense

Line 144 use ‘merged’ instead of ‘matched’

Line 146 use ‘excluded’ instead of ‘delated’

As there are further examples of language difficulties within the entire manuscript, I highly recommend having the manuscript reviewed by a native speaker before resubmission.

#2 It seems contradictory to me that Prof. Albert Nienhaus appears as a co-author of the manuscript (indicating his involvement in either study conduction, data analysis or interpretation) and also functions as head of the department for occupational medicine, hazardous substances and health sciences of the BGW (funder of the study). You state that ‘The funders had no role in study design, data collection and analysis, decision to publish, or preparation of the manuscript’. How does Prof. Nienhaus fulfill the criteria for authorship when he was not involved in any of these steps? Please clarify.

#3 Line 54 Do you mean information on palliative care qualifications that nurses have already absolved, or do you mean qualifications that can be obtained? Please clarify.

#4 Line 54/55 Why do you suppose that the degree of palliative care provision or the degree of obtained qualifications affect nurses’ health status? What is this hypothesis based on?

#5 Table 1 Why not include the footnote information in the main table? Leave out the last row with information on deceased persons per year as it is redundant information which has already been stated in the main text of the introduction section.

#6 Line 88 ‘Nurses’ health may have an effect on…..’ instead of ‘a nurse’s health’ (please change)

#7 Line 90-92 I suggest rephrasing your gap in research by highlighting the important role of moderator analysis and its’ benefits over e.g. correlation analysis. Then mentioning that the moderating role of palliative case aspects in burnout has not been investigated. This seems broader than stating that no moderator analysis has been done in the palliative care setting.

#8 Line 98 You speak of an explanatory study here but mention an exploratory study in lines 159 and 335. Please correct.

#9 Line 98 Please elaborate on the 10% sample of a database. What kind of database is it? What information does it contain? How was it collected and why? 10% of what?

#10 Line 101 Your response rate of 3.8% is very low. Please discuss possible reasons for the very low response rates and its implications for the validity of your findings in detail in the discussion section of your manuscript.

#11 Line 103 So the institutions and not the nurses chose whether online or paper-based questionnaires were distributed? Please clarify.

#12 Line 104 Of all 2,982 questionnaires how many were accessed online and how many were paper based?

#13 Line 106 Were questionnaires distributed to the nurses via their employers? This could introduce potential bias as employers may influence/push the nurses to give certain answers. Please discuss this in the discussion section of your manuscript.

#14 Line 116 Please clarify what is meant by ‘grade’.

#15 Lines 119-122 Please clarify if these items were self-developed and how they were developed (based on literature? Gut feeling?). If they were self-developed, please also discuss the validity of the items.

#16 Line 119/120 What exactly is meant by ‘number of patients’ deaths within the last month’? Does this refer to all dying patients within the institution or does it refer to the number of patients the nurse cared for personally? In the first case, the number would strongly depend on the size of the institution and would be the same for all nurses working in this institution.

#17 Line 139/140 Please rephrase the section ‘the nurses’ statement of resources in being helpful in dealing with the demands of their work’. I had to read this multiple times to understand.

#18 Line 142 What pilot study are you referring to? It has not been mentioned in any way before. What sample was involved? When and why was it conducted? Was it a qualitative or quantitative study?

#19 Line 163/164 What were the criteria for linear regression and what do you mean by ‘were treated as categorical variables’?

#20 Line 166 You decided to dichotomize the variable on the extent of palliative care. This is certainly legitimate. Yet, as the focus of your research paper is the analysis of palliative care aspects, it would have been interesting to look at this variable in greater details. Do nurses with even higher extent of palliative care feel more burnt out? Do nurses with an exceptionally low extent of palliative care not suffer from burnout?

#21 Line 169 change ‘per model 1’ into ‘one per model’ and add further clarification such as ‘Secondly, for all resources the following analysis was done:…..’ to make clear that the following methodology has been applied to all resources.

#22 Line 180 Start a new sentence when describing the response rate to avoid double brackets.

#23 Line 186/187 Always spell out numbers when they begin a sentence, e.g. ‘One hundred fifteen’ instead of 115.

#24 Table 2. What does the number 14 mean when describing professional experience? Years? Months?

#25 Table 2 and Table 4. Clarify what is meant by ‘yes + current qualification’ – does this mean that a participant is currently absolving further qualification? Then state so.

#26 Table 2. The number of missing values for the variable ‘extent of palliative care’ is exceptionally high. Why is this so? Please discuss this aspect in the discussion section as this item forms a major contribution to the novelty of your research.

#27 Table 3. All tables should be self-explanatory. Please add some reference to the COPSOQ questionnaire (as the table obviously refers to this instrument, yet this is not clarified).

#28 Line 201 I suppose the r-value is 0.498 instead of 498?

#29 Table 4. You developed the list of personal and social resources after conduction of a pilot study as you stated. Have you conducted any analysis of discriminatory power of these items? E.g. analyzing the distinction between ‘hobbies’ and ‘sport’ because – at least to me – sport is a common hobby.

#30 Line 240 Do you really mean -0.34 instead of -34? This is indeed a very small effect and is only shortly discussed in lines 270/271. Please elaborate more on why the effect is so small.

#31 Line 280/281 What occupational group does this reference refer to?

#32 Within the methods section of your manuscript you state ‘resources that reached a p-value <0.05 in the bivariate analysis with the scale ‘burnout’ were further analysed in the moderator analysis’. In lines 287-289 you state that several resources were significantly associated with burnout. Yet, why was no moderator analysis done for these resources? Or why did you decide not to depict results? Please clarify.

#33 Line 308/309 ‘We observed no influence of an additional qualification on the quantitative demand- burnout relationship’ – where was this analysis done? Where do you depict results?

#34 Line 322/323 Do you hold any information on the number of palliative care patients per institution? This would have been interesting to know.

#35 Figure 2. For low quantitative demands, why do you observe higher burnout rates among participants who report a good working team? Please discuss.

Reviewer #3: The study reports findings from a cross-sectional survey that investigates the moderating effects of resources on the workload - burnout association among nurses in non-specialized palliative care. In my view, there is an ongoing need for mental health studies in the nursing sector. As such, the study addresses a relevant and timely topic. Nevertheless, I see shortcomings in the clarity of the contribution, the incorporation of relevant research literature, the transparency of the analyses as well as the justification of some conclusions. I hope my comments can help to further improve the study.

1. Please better justify your study approach by explaining why the results of previous studies cannot be directly transferred to non-specialized palliative care.

2. I would not agree with your statement on page 4: “ Studies examining the buffering/moderating role of resources on the relationship between workload and burnout are rare.” Please consider for example the findings on the Job Demand-Resources Model (e.g. Demerouti et al., 2001), which has been dealing with this question for quite some time. (Demerouti, E., Bakker, A. B., Nachreiner, F., & Schaufeli, W. B. (2001). The Job Demands-Resources Model of Burnout. Journal of Applied Psychology, 56(3), 499-512.)

3. Please provide in your introduction a clear definition of “resources”, as well as personal, social and organizational resources. Please also explain why and how these three kinds of resources should moderate the association between workload and burnout.

4. Please report in the method-section the value range of the COPSOQ-Scales. This would facilitate the interpretation of your mean values.

5. Regarding your analyses and findings on the moderation-effects (Table 5), it is not quite clear to me why you report specifically these three interaction effects. Please clarify, if there where theoretical considerations to test only these three interaction effects, or did your report only the significant interactions. In this latter case I would suggest to report and discuss also the non-significant findings.

6. Since the requirements in palliative care are in the focus of this study, it would be desirable to know how resources moderate the effects of these specific demands (e.g. extent of palliative care) on burnout. At the moment we only learn from the study in respect to palliative care that the extent of palliative care has an additive main effect beyond more general work demands in nursing.

7. In your discussion of the buffering effects of commitment, you compare your findings with the findings of other studies that - in contrast to your study - seem to have examined only the direct effects of commitment on burnout. I think that your study cannot be straightforwardly compared with these studies because both are based on different model assumptions. Thus your statement, that the moderation analyses is a specific strength of your study (page 15) is not plausible in my opinion. Instead, it would be more enlightening if you could go into more detail about possible different mechanisms that can explain the direct or moderating effects of commitment in relation to burnout.

8. On page 15 (and similarly in the conclusions) you state that “It can be assumed that good collaboration within the team and supervisors stimulates workplace commitment.” In my opinion this conclusion cannot be deduced from your findings. Please report results that support this conclusion or revise this statement.

9. Line 295-298 on page 16 should be moved to the limitations.

6. PLOS authors have the option to publish the peer review history of their article (what does this mean?). If published, this will include your full peer review and any attached files.

Reviewer #1: No

Reviewer #2: No

Reviewer #3: No

---

## [Author Response · Author response to Decision Letter 0]

6 Nov 2020

Responses to the academic editor

We note that you have indicated that data from this study are available upon request. PLOS only allows data to be available upon request if there are legal or ethical restrictions on sharing data publicly. For information on unacceptable data access restrictions, please see http://journals.plos.org/plosone/s/data-availability#loc-unacceptable-data-access-restrictions.

Response: According to the Ethics Committee of the Medical Association of Rhineland-Palatinate (Study ID: 837.326.16 (10645)), the Institute of Occupational, Social and Environmental Medicine of the University Medical Center of the University Mainz is specified as data holding organization. The institution is not allowed to share the data publically in order to guarantee anonymity to the institutions that participated in the survey because some institution-specific information could be linked to specific institutions. The data set of the present study is stored on the institution server at the University Medical Centre of the University of Mainz and can be requested for scientific purposes via the institution office. This ensures that data will be accessible even if the authors of the present paper change affiliation. Postal address: University Medical Center of the University of Mainz, Institute of Occupational, Social and Environmental Medicine, Obere Zahlbacher Str. 67, D-55131 Mainz. Email address: arbeitsmedizin@uni-mainz.de

Response: Thank you very much. We will update our Data Availability statement in the submission system. 

Thank you for stating the following in the Competing Interests section:

'I have read the journal's policy and the authors of this manuscript have the following competing interests: The project was funded by the BGW - Berufsgenossenschaft für Gesundheitsdienst und Wohlfahrtspflege (Institution for Statutory Accident Insurance and Prevention in Health and Welfare Services). The BGW is responsible for the health concerns of the target group investigated in the present study, namely nurses. Prof. Dr. A. Nienhaus is head of the Department for Occupational Medicine, Hazardous Substances and Health Science of the BGW and co-author of this publication. All other authors declare to have no potential conflict of interest. '

a. Please confirm that this does not alter your adherence to all PLOS ONE policies on sharing data and materials, by including the following statement: "This does not alter our adherence to PLOS ONE policies on sharing data and materials.” (as detailed online in our guide for authors http://journals.plos.org/plosone/s/competing-interests). If there are restrictions on sharing of data and/or materials, please state these.

Please note that we cannot proceed with consideration of your article until this information has been declared.

Response: We update our Competing Interests statement as follows:

'I have read the journal's policy and the authors of this manuscript have the following competing interests: The project was funded by the BGW - Berufsgenossenschaft für Gesundheitsdienst und Wohlfahrtspflege (Institution for Statutory Accident Insurance and Prevention in Health and Welfare Services). The BGW is responsible for the health concerns of the target group investigated in the present study, namely nurses. Prof. Dr. A. Nienhaus is head of the Department for Occupational Medicine, Hazardous Substances and Health Science of the BGW and co-author of this publication. All other authors declare to have no potential conflict of interest. This does not alter our adherence to PLOS ONE policies on sharing data and materials. '

Responses to Reviewer 1

Overall comments: Congratulations to the authors for their work. It is a relevant topic and has an impact on clinical practice.

The limitations that I identified in the studies were assumed by the authors in the limitations section.

In the methodological part, although the tests used are perceived, it was not clear to me the assumptions behind them (parametric studies vs. nonparametric studies), how this option was assumed.

The discussion can be further developed to highlight the results obtained in the light of the existing evidence.

Response to overall comments: Thank you very much for the positive words on our manuscript and your constructive comments. We significantly revised parts of the methods and discussion sections aiming to strengthen traceability of the methodological procedure and to improve intelligibility. We hope that you will be satisfied with the revised version. If you should have any further recommendations for improving our manuscript, please don´t hesitate to communicate these to us. 

Responses to Reviewer 2

Overall comments: The present study deals with burnout in the context of nursing and the possible moderating role of several personal, social, and organizational resources on the relationship between quantitative demands and burnout. It is the first study to also include palliative care aspects such as the ‚extent of palliative care‘ within different nursing tasks and the ‚number of patient deaths during the last month‘. The results of the study indicate a moderating role of workplace commitment, a good working team and recognition of one’s supervisor on the relationship between quantitative demands and burnout among nurses.

A very comprehensive questionnaire is used including many validated scales such as parts of the Copenhagen Psychosocial Questionnaire and the RS-13. The study design seems appropriate in order to answer the research question.

Yet, some major limitations apply. All in all, the manuscript is somewhat confusing to read, which is not least due to many linguistic irregularities, but also to a partly imprecise and sketchy presentation of the methodology. Furthermore, the study sample is not representative for nurses working in palliative care in Germany and the response rate of 3.8% is exceptionally low.

Response to overall comments: Thank you very much for the positive words on our manuscript and your constructive comments. We are grateful for the detailed suggestions that were very helpful for improving our manuscript. 

With regard to your suggestions, we significantly revised the methodology section of the manuscript (lines 193-199) to make it clearer for the reader. We are aware of the low response rate and we highlight and discuss this in the limitation section of the manuscript recommending future studies with larger samples. However, we think it is important for the specific field of palliative care to report the findings mentioned in the paper. In order to achieve a better understanding on how the study sample was recruited and which challenges occur when surveying nurses in non-specialised palliative care settings, we revised the manuscript sections on study design and participants (lines 113-126). We hope that you will be satisfied with the revised version in which we have incorporated your points. If you should have any further recommendations for improving our manuscript, please communicate these to us. Please see the manuscript with Track Changes. 

Specific comments:

Comment 1: I assume that the manuscript has not been reviewed by a native speaker. Both linguistic oddities and incorrect use of grammar and verbs are noticeable. Examples:

Line 66 possessive apostrophe missing (it should be nurses’ instead of nurses)

Line 68/69 use ‘alarming increase’ instead of ‘worrying development’

Line 71 s is missing (it should be concerns instead of concern)

Line 77/78 use ‘such as’ or ‘e.g.’ instead of ‘like’

Line 137 sudden change from past tense to present tense

Line 144 use ‘merged’ instead of ‘matched’

Line 146 use ‘excluded’ instead of ‘delated’

As there are further examples of language difficulties within the entire manuscript, I highly recommend having the manuscript reviewed by a native speaker before resubmission.

Response 1: Thank you very much for the examples. We fully agree and following your suggestion, the whole manuscript was proofread by a native speaker with a scientific background. 

Comment 2: It seems contradictory to me that Prof. Albert Nienhaus appears as a co-author of the manuscript (indicating his involvement in either study conduction, data analysis or interpretation) and also functions as head of the department for occupational medicine, hazardous substances and health sciences of the BGW (funder of the study). You state that ‘The funders had no role in study design, data collection and analysis, decision to publish, or preparation of the manuscript’. How does Prof. Nienhaus fulfill the criteria for authorship when he was not involved in any of these steps? Please clarify.

Response 2: Thank you very much for this comment. Albert Nienhaus is head of the department of occupational medicine, hazardous substances and health sciences of the BGW and head of the the center for epidemiology and health service research in nursing (CVcare) of the University Clinics in Hamburg Eppendorf (UKE). In his function as head of the department of occupational medicine, hazardous substances and health sciences of the BGW, he helped to prepare the research proposal for a research grant to be obtained from the BGW. The self-government of the BGW decided to support the study financially. However, the governmental body of the BGW did not influence data analysis, data interpretation or the decision to publish. In his function as head of the CVcare, Prof. Nienhaus was engaged in developing the study design, data collection and preparation of the manuscript. The work of the CVcare is sponsored by the BGW. As part of the German social security system, the BGW is a non-profit organisation which has the legal obligation to promotes safety and health at the workplace via supporting independent research. 

Thus, we would like to leave the Financial Disclosure statement ‘The funder had no role in study design, data collection and analysis, decision to publish, or preparation of the manuscript’ as it is. 

Comment 3: Line 54 Do you mean information on palliative care qualifications that nurses have already absolved, or do you mean qualifications that can be obtained? Please clarify.

Response 3: Thank you for this comment. We mean palliative care qualifications that nurses have already absolved. For clarification, we revised the sentence (line 57). 

Comment 4: Line 54/55 Why do you suppose that the degree of palliative care provision or the degree of obtained qualifications affect nurses’ health status? What is this hypothesis based on?

Response 4: The assumption that the degree of palliative care provision or the degree of obtained qualifications affect nurses’ health status was based on a literature search (we incorporated new text into the manuscript line 58-60) and the results of the previous pilot study, which was conducted in specialized palliative care. For more information, please see lines 142-145, where we go into detail with the pilot study. Within the pilot study we obtained a positive effect of the additional palliative care qualification in relation to organisational demands and demands regarding the care of relatives (lines 368-371). 

Comment 5: Table 1 Why not include the footnote information in the main table? Leave out the last row with information on deceased persons per year as it is redundant information which has already been stated in the main text of the introduction section.

Response 5: Following your comment, we deleted the last row of the table. But we would like to leave the footnote information as footnote, because the facilities named in the table are the facilities which account for the largest part of palliative care in Germany and which were investigated by our study team. The facilities named in the footnotes are only mentioned for the sake of completeness. 

Comment 6: Line 88 ‘Nurses’ health may have an effect on…..’ instead of ‘a nurse’s health’ (please change)

Response 6: Following your comment, we revised the sentence (line 103).

Comment 7: Line 90-92 I suggest rephrasing your gap in research by highlighting the important role of moderator analysis and its’ benefits over e.g. correlation analysis. Then mentioning that the moderating role of palliative case aspects in burnout has not been investigated. This seems broader than stating that no moderator analysis has been done in the palliative care setting.

Response 7: Thank you very much for your suggestion. Please see lines 88-93, where we point out the benefits of a moderator analysis. Because the research gap focuses in particular on the consideration of palliative care aspects and not on moderator analysis, we revised this sentence (lines 106-107). 

Comment 8: Line 98 You speak of an explanatory study here but mention an exploratory study in lines 159 and 335. Please correct.

Response 8: Thank you very much for this comment, we corrected the sentence (line 113).

Comment 9: Line 98 Please elaborate on the 10% sample of a database. What kind of database is it? What information does it contain? How was it collected and why? 10% of what?

Response: According to your comment, we revised the sections on study design and participants in order to clarify our sampling procedure (lines 112-126). 

Comment 10: Line 101 Your response rate of 3.8% is very low. Please discuss possible reasons for the very low response rates and its implications for the validity of your findings in detail in the discussion section of your manuscript.

Response 10: Please find our critical reflection of this point in the limitations section of the manuscript (lines 379-385) where we state that the results of the present study may be labelled as preliminary due to low participation rate.

Comment 11: Line 103 So the institutions and not the nurses chose whether online or paper-based questionnaires were distributed? Please clarify.

Response 11: No, the nurses decided. In a first step, the study team got access to the addresses of all institutions which agreed to participate in the study. Then, all institutions were personally contacted by the study team and asked, how many nurses would be working there and whether the nurses would prefer to prepare a paper-and-pencil questionnaire or an online survey. To clarify this point, we revised the respective part in the manuscript (line 123). 

Comment 12: Line 104 Of all 2,982 questionnaires how many were accessed online and how many were paper based?

Response 13: In order to give the reader a detailed overview on this aspect, we now added an additional table (additional Table 1) to the manuscript which show the number of questionnaires sent out to the facilities (paper or online) and the response rates (line 126). 

Comment 13: Line 106 Were questionnaires distributed to the nurses via their employers? This could introduce potential bias as employers may influence/push the nurses to give certain answers. Please discuss this in the discussion section of your manuscript.

Response 13: Yes, the questionnaires were distributed to the nurses via their employers. But as we wrote in the study design and participants section, participation was voluntary and anonymous. Every nurse received either a paper-and-pencil questionnaire with a pre-franked envelope or an access code to the online survey. The paper-and-pencil version potentially leaves room for the employers to influence the nurses’ answers. However, we assume that the employers and the nurses, which have agreed to participate in this anonymous and voluntary survey, did this in accordance to the survey introduction. We already discuss potential biases in the limitations section (lines 379-389) and would appreciate not to make a potential failure of the employers and nurses to a subject of the manuscript.

Comment 14: Line 116 Please clarify what is meant by ‘grade’.

Response 14: Thank you very much for this comment. We replaced the word “grade” by the word “professional qualification” (line 136, Table 2, line 226). 

Comment 15: Lines 119-122 Please clarify if these items were self-developed and how they were developed (based on literature? Gut feeling?). If they were self-developed, please also discuss the validity of the items.

Response 15: Yes, these items were self-developed. The first two items were already used in the pilot study. The pilot study consisted of a qualitative part, where interviews with experts in general and specialised palliative care were conducted. These interviews were used to develop a standardized questionnaire which was used for a cross-sectional survey. We now included this information into the manuscript (lines 139-145). We also added a sentence to the validity of these items to the limitations section (lines 389-391). 

Comment 16: Line 119/120 What exactly is meant by ‘number of patients’ deaths within the last month’? Does this refer to all dying patients within the institution or does it refer to the number of patients the nurse cared for personally? In the first case, the number would strongly depend on the size of the institution and would be the same for all nurses working in this institution.

Response 16: Thank you very much for this comment. This question refers to the number of patients the nurses cared for personally. To make this point clear, we added additional information to the manuscript (line 140).

Comment17: Line 139/140 Please rephrase the section ‘the nurses’ statement of resources in being helpful in dealing with the demands of their work’. I had to read this multiple times to understand.

Response 17: Following your comment, we revised this section for improving readability (line 162-165).

Comment 18: Line 142 What pilot study are you referring to? It has not been mentioned in any way before. What sample was involved? When and why was it conducted? Was it a qualitative or quantitative study?

Response 18: The pilot study consisted of a qualitative and a quantitative part. We now added a description of the pilot study as well as the referring literature into the manuscript (lines 143-145).

Comment 19: Line 163/164 What were the criteria for linear regression and what do you mean by ‘were treated as categorical variables’?

Response 19: We performed regression based moderation analysis. Moderation analysis involves the use of linear or logistic multiple regression analysis. Therefore, we first checked the criteria for linear (e.g. linearity, normally distributed errors) and logistic regression (e.g. linearity between continuous predictors and the logit of the outcome variable) according to Reference 35 (Field A. Discovering statistics using IBM SPSS statistics. 4th ed. Los Angeles, London, New Delhi, Singapore, Washington DC, Melbourne: SAGE; 2016). The variable age for example did not fulfil the conditions. Therefore we recoded the metric variable age into a categorical variable with three groups for the moderation analysis (see table 5). We revised the sentence in ‘were recoded as categorical variables’ (line 190). 

Comment 20: Line 166 You decided to dichotomize the variable on the extent of palliative care. This is certainly legitimate. Yet, as the focus of your research paper is the analysis of palliative care aspects, it would have been interesting to look at this variable in greater details. Do nurses with even higher extent of palliative care feel more burnt out? Do nurses with an exceptionally low extent of palliative care not suffer from burnout?

Response 20: You mention a very interesting point. Following your suggestion, we revised the descriptive results section by including further results with respect to the variable extent of palliative care (lines 217-221). Regarding the relationship between the extent of palliative care and burnout, please see lines 237-239 in the bivariate analyses section. We could also report, that the extent of palliative care was positively correlated with the burnout score (r(314) = .153, p = .007). But because the extent of palliative care is not normally distributed, we don’t think that there is a big knowledge gain in doing this. The questions you asked above are very interesting, but as it would be too little to only analyse the relationship between the extent of palliative care and burnout without other variables (like quantitative demands, emotional demands) we would not like to go further into detail. 

Comment 21: Line 169 change ‘per model 1’ into ‘one per model’ and add further clarification such as ‘Secondly, for all resources the following analysis was done:…..’ to make clear that the following methodology has been applied to all resources.

Response 21: Following your suggestion, we changed the sentence (lines 199) and revised parts of the manuscript in order to clarify our statistical analysis plan (lines 193-199). 

Comment 22: Line 180 Start a new sentence when describing the response rate to avoid double brackets.

Response 22: Thank you very much for this comment, which we followed (lines 210-211).

Comment 23: Line 186/187 Always spell out numbers when they begin a sentence, e.g. ‘One hundred fifteen’ instead of 115.

Response 23: Thank you very much for this remark. We revised the sentences in line 217 and line 221.

Comment 24: Table 2. What does the number 14 mean when describing professional experience? Years? Months?

Response 24: Thank you very much for this comment, it means 14 years. Table 2 was revised.

Comment 25: Table 2 and Table 4. Clarify what is meant by ‘yes + current qualification’ – does this mean that a participant is currently absolving further qualification? Then state so.

Response 25: Yes, this means that a participant is currently absolving further qualification. This information was added to table 2 and table 4.

Comment 26: Table 2. The number of missing values for the variable ‘extent of palliative care’ is exceptionally high. Why is this so? Please discuss this aspect in the discussion section as this item forms a major contribution to the novelty of your research.

Response: Thank you very much for this comment. We incorporated a paragraph at the end of the discussion section addressing this important aspect (lines 374-377).

Comment 27: Table 3. All tables should be self-explanatory. Please add some reference to the COPSOQ questionnaire (as the table obviously refers to this instrument, yet this is not clarified).

Response 27: Following your comment, we revised the title of Table 3 (line 231).

Comment 28: Line 201 I suppose the r-value is 0.498 instead of 498?

Response 28: Thank you very much for reading our paper so carefully. Yes the r-value is 0.498, it was corrected in the manuscript (line 235).

Comment 29: Table 4. You developed the list of personal and social resources after conduction of a pilot study as you stated. Have you conducted any analysis of discriminatory power of these items? E.g. analyzing the distinction between ‘hobbies’ and ‘sport’ because – at least to me – sport is a common hobby.

Response 30: You mention an important point. An analysis of discriminatory power was done for the items which were made to scales. As the variables ‘hobbies’ and ‘sport’ has different frequency distributions and were not made to scales, no such analysis was performed. 

Comment 30: Line 240 Do you really mean -0.34 instead of -34? This is indeed a very small effect and is only shortly discussed in lines 270/271. Please elaborate more on why the effect is so small.

Response: Yes, the result of the analysis is really -0.34. But although the effect is small, to our opinion it is an interesting result which is consistent with the findings of previous studies. Please see lines 391-397 of the limitations section, where we discuss the limitations of the results of the moderation analysis.

Comment 31: Line 280/281 What occupational group does this reference refer to?

Response 31: Thank you very much for this comment. We incorporated the occupational groups not only to this reference but also to the two following references in the manuscript (line 314, lines 320-321, line 327, line 328, line 331).

Comment 32: Within the methods section of your manuscript you state ‘resources that reached a p-value <0.05 in the bivariate analysis with the scale ‘burnout’ were further analysed in the moderator analysis’. In lines 287-289 you state that several resources were significantly associated with burnout. Yet, why was no moderator analysis done for these resources? Or why did you decide not to depict results? Please clarify.

Response 32: Thank you very much for this comment. As Reviewer 3 had comparable comments on our analysis plan, we revised the whole section aiming to improve traceability. Please see lines 193-199 in the data preparation an analysis section as well as our changes in the results section where the moderator analysis is presented (lines 254-255). As we now describe in more detail, a moderation analysis was done for all resources which were significantly associated with burnout, but we only report the resources which significantly moderated burnout. In total, we conducted 16 moderation analyses but we decided not to report the data of the non-significant moderations, because this would mean to present an enormous mass of data that will not lead to an extensive gain of knowledge. The resources which were significant within the bivariate analysis but did not moderate burnout are discussed in the discussion section (lines 344-356).

Comment 33: Line 308/309 ‘We observed no influence of an additional qualification on the quantitative demand- burnout relationship’ – where was this analysis done? Where do you depict results?

Response 33: You are absolutely right, we did no moderation analysis of an ‘additional qualification on the quantitative demand- burnout relationship’ because the additional qualification was not significantly associated with burnout (Table 4). We revised the sentence according to the results of the study (lines 366-368).

Comment 34: Line 322/323 Do you hold any information on the number of palliative care patients per institution? This would have been interesting to know.

Response: You are right, this would have been interesting to know. However, we did not address this issue in the survey. Furthermore, for reasons of anonymity, we were not able to assign single questionnaires to specific institutions and therefore, we were not able to objectively assess the number of patients in general and palliative patients per institution, for example by asking the facility managers or head nurses. But we will consider this aspect in our future studies. 

#35 Figure 2. For low quantitative demands, why do you observe higher burnout rates among participants who report a good working team? Please discuss.

Response: Thank you very much for this comment. We incorporated this aspect to the discussion section (lines 335-340). 

Responses to Reviewer 3

Overall comment: The study reports findings from a cross-sectional survey that investigates the moderating effects of resources on the workload - burnout association among nurses in non-specialized palliative care. In my view, there is an ongoing need for mental health studies in the nursing sector. As such, the study addresses a relevant and timely topic. Nevertheless, I see shortcomings in the clarity of the contribution, the incorporation of relevant research literature, the transparency of the analyses as well as the justification of some conclusions. I hope my comments can help to further improve the study.

Response to overall comment: Thank you very much for the positive words on our manuscript and your constructive comments. We are grateful for the detailed suggestions that were very helpful for improving our manuscript. 

We hope that you will be satisfied with the revised version in which we have incorporated your points. If you should have any further recommendations for improving our manuscript, please communicate these to us. 

Comment 1: Please better justify your study approach by explaining why the results of previous studies cannot be directly transferred to non-specialized palliative care.

Response 1: Thank you very much for this comment. We revised the introduction section, in order to better justify why the results of previous studies cannot be directly transferred to non-specialized palliative care (lines 49-51, lines 56-57, lines 65-66). Please see the manuscript with Track Changes. 

Comment 2: I would not agree with your statement on page 4: “ Studies examining the buffering/moderating role of resources on the relationship between workload and burnout are rare.” Please consider for example the findings on the Job Demand-Resources Model (e.g. Demerouti et al., 2001), which has been dealing with this question for quite some time. (Demerouti, E., Bakker, A. B., Nachreiner, F., & Schaufeli, W. B. (2001). The Job Demands-Resources Model of Burnout. Journal of Applied Psychology, 56(3), 499-512.)

Response 2: We fully agree with you. We revised this sentence (lines 89-91) as the word “rare” referred rather to studies examining the moderating role of resources in nursing than to the statistical approach. On the basis of the reference you have provided, we found another paper using moderator analysis in nursing, which we added to the manuscript (Reference 38, Xanthopoulou, D., Bakker, A. B., Dollard, M. F., Demerouti, E., Schaufeli, W. B., Taris, T. W., & Schreurs, P. J. (2007). When do job demands particularly predict burnout?: The moderating role of job resources. Journal of Managerial Psychology, 22(8), 766-786). We think we missed this paper, because it focuses on home care organization employees and our literature search concentrated especially on nurses. 

Comment 3: Please provide in your introduction a clear definition of “resources”, as well as personal, social and organizational resources. Please also explain why and how these three kinds of resources should moderate the association between workload and burnout.

Response 3: Thank you very much for this comment. We really understand your point as you are missing a concrete model, like for example the Job Demand-Resources Model, to underpin this study. Since the present study was conducted in the framework of a doctoral thesis in occupational science in Germany, the theoretical basis of this study was Rudow’s Stress-Strain-Resources model. This model is an extension of the basic model of stress and strain in work science in Germany, which was originally developed by Rohmert and Rutenfranz (1983) and the concept of Salutogenesis by Antonovsky. According to Rudow, individual, social and organisational resources of a person buffer/moderate the negative effects of job demands (stress) on health (strain) [Rudow, B. (2004). Das gesunde Unternehmen. Gesundheitsmanagement, Arbeitsschutz und Personalpflege in Organisationen. München & Wien: Oldenbourg]. In addition, the model describes that stress can lead to different strains in different people depending on available resources (e.g. team support or persons personal capacities like qualification). This resources can be either individual, social or organisational and buffer/moderate the negative effects of job demands (stress) on, for example, burnout (strain). The individual, social and organisational resources which were investigated in the present study are based on a previously performed pilot study, which we present in more detail in lines 143-145 of the manuscript. According to the results of this pilot study, these resources were of high importance by nurses in specialised palliative care. Since our pilot study as well as Rudow’s model has only been published in German language, we first decided not to mention these in the paper. Furthermore, a detailed presentation of the model in detail would go beyond the scope of this manuscript. Please see lines 98-102 where we now added the most important aspects of Rudow’s Stress-Strain-Resources model to the manuscript.

Comment 4: Please report in the method-section the value range of the COPSOQ-Scales. This would facilitate the interpretation of your mean values.

Response 4: Following your suggestion, we added this information to table 3.

Comment 5: Regarding your analyses and findings on the moderation-effects (Table 5), it is not quite clear to me why you report specifically these three interaction effects. Please clarify, if there where theoretical considerations to test only these three interaction effects, or did your report only the significant interactions. In this latter case I would suggest to report and discuss also the non-significant findings.

Response 5: As Reviewer 2 had comparable comments on our analysis plan, we revised this section aiming to improve traceability. Therefore, please see our changes in lines 193-199 in the data preparation an analysis section as well as the changes in the results section where the moderator analyses are presented (lines 254-255). As we now describe in more detail, a moderation analysis was done for all resources which were significantly associated with burnout, but we only report the resources which significantly moderated burnout. In total, we conducted 16 moderation analyses but we decided not to report the data of the non-significant moderations, because this would mean to present an enormous mass of data that will not lead to an extensive gain of knowledge. The resources which were significant within the bivariate analysis but did not moderate burnout are discussed in the discussion section (lines 344-356).

Comment 6: Since the requirements in palliative care are in the focus of this study, it would be desirable to know how resources moderate the effects of these specific demands (e.g. extent of palliative care) on burnout. At the moment we only learn from the study in respect to palliative care that the extent of palliative care has an additive main effect beyond more general work demands in nursing.

Response 6: Thank you very much for this comment. This shows us, how important the focus on palliative care is. At the moment, we are preparing a manuscript addressing the palliative care requirements (e.g. extent of palliative care) in relation to other occupational demands (not only quantitative demands but also burden due to organisational framework conditions, emotional demands, demands for hiding emotions, emotional burden due to death, burden due to care of patients, burden due to nursing care and burden due to care of relatives) and their association with resources, health, and intention to leave the profession of nurses. Putting all this information into one paper would go beyond the scope of the manuscript. 

Comment 7: In your discussion of the buffering effects of commitment, you compare your findings with the findings of other studies that - in contrast to your study - seem to have examined only the direct effects of commitment on burnout. I think that your study cannot be straightforwardly compared with these studies because both are based on different model assumptions. Thus your statement, that the moderation analyses is a specific strength of your study (page 15) is not plausible in my opinion. Instead, it would be more enlightening if you could go into more detail about possible different mechanisms that can explain the direct or moderating effects of commitment in relation to burnout.

Response 7: Following your comment, we revised this paragraph (lines 313-334). 

Comment 8: On page 15 (and similarly in the conclusions) you state that “It can be assumed that good collaboration within the team and supervisors stimulates workplace commitment.” In my opinion this conclusion cannot be deduced from your findings. Please report results that support this conclusion or revise this statement.

Response 8: Following your comment, we revised this sentence by referring to the results of other studies analysing workplace commitment (lines 313-334). 

Comment 9: Line 295-298 on page 16 should be moved to the limitations.

Response 9: Thank you very much for this comment. We deleted the first part of this sentence in the discussion section and incorporated it to the limitations section (lines 351-352 and line 397-401).

---

## [Decision Letter · Decision Letter 1]

9 Dec 2020

PONE-D-20-23720R1

The relationship between workload and burnout among nurses in non-specialized palliative care settings: the buffering role of personal, social and organisational resources

PLOS ONE

Dear Dr. Dietz,

Thank you for submitting your manuscript to PLOS ONE. After careful consideration, we feel that it has merit but does not fully meet PLOS ONE’s publication criteria as it currently stands. Therefore, we invite you to submit a revised version of the manuscript that addresses the points raised during the review process.

We look forward to receiving your revised manuscript.

Kind regards,

Adrian Loerbroks

Academic Editor

PLOS ONE

Reviewers' comments:

Reviewer's Responses to Questions

**Comments to the Author**

1. If the authors have adequately addressed your comments raised in a previous round of review and you feel that this manuscript is now acceptable for publication, you may indicate that here to bypass the “Comments to the Author” section, enter your conflict of interest statement in the “Confidential to Editor” section, and submit your "Accept" recommendation.

Reviewer #1: All comments have been addressed

Reviewer #2: All comments have been addressed

Reviewer #3: (No Response)

2. Is the manuscript technically sound, and do the data support the conclusions?

Reviewer #1: Yes

Reviewer #2: Yes

Reviewer #3: Partly

3. Has the statistical analysis been performed appropriately and rigorously? 

Reviewer #1: Yes

Reviewer #2: Yes

Reviewer #3: No

4. Have the authors made all data underlying the findings in their manuscript fully available?

Reviewer #1: Yes

Reviewer #2: No

Reviewer #3: No

5. Is the manuscript presented in an intelligible fashion and written in standard English?

Reviewer #1: Yes

Reviewer #2: Yes

Reviewer #3: No

6. Review Comments to the Author

Reviewer #1: The authors made the requested changes. The present manuscript presents another quality of information and clarity.

Reviewer #2: The authors have provided a profound revision and have adequately adressed all my comments. It becomes clear that the manuscript has been proof-read by a native English speaker. I especially liked the revised introduction section that clearly states the difference in qualification and time spent per patient between nurses in specialised and general palliative care settings. The manuscript additionally gained in transparency by e.g. giving more details on the conducted pilot study or by describing the amount of pallitative care among the respondents in more detail. All tables have been adapted adequately.

Some minor comments are to be found below:

- Table 1. If the institutions mentioned in the footnote were not included in the study, I suggest to state this so that it becomes clear why certain institutions were placed in the main table and some in the footnote.

- Line 97: I personally would not mention details of methodology in the introduction section. It seems enough to just mention the presence of the model that was later used.

- Line 144: I consider the word "subgroup" confusing here. Subgroup of what? Simply state that the database contained outpatient facilities, hospitals and nursing homes

- Lines 215/218: Percentages do not agree for amount of nurses that provide palliative care in >20% of their working time. Line 215 states 36.4%, line 218 states 26.3%. Please correct.

- Lines 373-376. I don’t think that making palliative care to a subject of discussion solely justifies the high amount of missing data for the key variable of interest.

- Line 381: Say "exclusion of hospitals" instead of "exclusion from hospitals"

- Line 393: Say "additional amount" instead of "additionally amount"

Reviewer #3: I thank the authors for implementing my comments. However, I see room for further improvement on some points:

1. Thanks for your additional explanations on your theoretical model. However, I still miss a definition what a resource is an why it may buffer the effects of work demands on strain.

2. A note on the terms "stress and strain". In my opinion stress, like strain, is an individual reaction to an external demand or stressor (see Transactional Stress Model of Lazarus or Selyes adaptation syndrome). I therefore suggest using the term stressor instead of stress.

3. All in all, it would be good to check once again that the main terms are used consistently. Especially workload and quantitative demands are used interchangeably.

4. It is not plausible why you only included those resources as moderators that showed statistically significant bivariate correlations with burnout, as this is not a prerequisite (neither statistical nor conceptual) for moderator analyses. Since your approach is explorative, I suggest that you use all resources of moderator analysis.

5. Your explanation for not reporting interactions between specific demands in palliative care and resources is not very satisfying to me, as your study aims “to investigate the buffering role of resources on the relationship between workload and burnout among nurses in non-specialized palliative care settings, with consideration given to palliative care aspects, such as information on the ‘extent of palliative care’” (p.5) In this sense, your analyses do not completely meet the objective of your study.

7. PLOS authors have the option to publish the peer review history of their article (what does this mean?). If published, this will include your full peer review and any attached files.

Reviewer #1: No

Reviewer #2: No

Reviewer #3: No

---

## [Author Response · Author response to Decision Letter 1]

15 Dec 2020

Responses to Reviewer 1

Reviewer #1: The authors made the requested changes. The present manuscript presents another quality of information and clarity.

Response: Thank you very much for the positive words on our manuscript and your constructive comments which were very helpful for improving our manuscript. We are happy that you find our paper suitable for publication in PLoS One.

Responses to Reviewer 2

Reviewer #2: The authors have provided a profound revision and have adequately addressed all my comments. It becomes clear that the manuscript has been proof-read by a native English speaker. I especially liked the revised introduction section that clearly states the difference in qualification and time spent per patient between nurses in specialised and general palliative care settings. The manuscript additionally gained in transparency by e.g. giving more details on the conducted pilot study or by describing the amount of palliative care among the respondents in more detail. All tables have been adapted adequately.

Response to overall comments: Thank you very much for the positive words on our manuscript and for the additional comments for further improving our manuscript. We hope that you will be satisfied with the revised version in which we have incorporated your points. If you should have any further recommendations, please don´t hesitate to communicate these to us. 

Response to your comments:

Comment 1: Table 1. If the institutions mentioned in the footnote were not included in the study, I suggest to state this so that it becomes clear why certain institutions were placed in the main table and some in the footnote.

Response 1: Thank you very much for this comment. For clarification, we added a new sentence to the footnote (line 68-69). 

Comment 2: Line 97: I personally would not mention details of methodology in the introduction section. It seems enough to just mention the presence of the model that was later used.

Response 2: Thank you very much for this comment. We incorporated some details on methodology into the introduction because Reviewer 3 missed this information in the introduction. Since Reviewer 3 also had additional comments (see below) on our manuscript, we would like to leave this information in the introduction. We hope for your understanding. 

Comment 3: Line 144: I consider the word "subgroup" confusing here. Subgroup of what? Simply state that the database contained outpatient facilities, hospitals and nursing homes

Response 3: Following your comment, we revised the sentence (line 117).

Comment 4: Lines 215/218: Percentages do not agree for amount of nurses that provide palliative care in >20% of their working time. Line 215 states 36.4%, line 218 states 26.3%. Please correct.

Response 4: Thank you very much for your careful reading and this comment identifying a mistake. 36.4% is correct. The relative percentages in lines 216-219 came from an old manuscript version and included the nurses which did not answer this specific question. We apologize for this mistake. We now checked and corrected the numbers in the manuscript and added additional information to the manuscript (line 217-221, line 223) in order to clarify this point.

Comment 5: Lines 373-376. I don’t think that making palliative care to a subject of discussion solely justifies the high amount of missing data for the key variable of interest.

Response 5: Thank you very much for this comment. This is not what we wanted to say. We think that the main contribution of the present study is to make palliative care aspects in non-specialised palliative care settings a subject of discussion regardless of the amount of missing data for the variable. For clarification, we incorporated an additional paragraph into the manuscript and revised the last sentence of the discussion section (line 367).

Comment 6: Line 381: Say "exclusion of hospitals" instead of "exclusion from hospitals"

Response 6: Following your comment, we revised the sentence (line 373).

Comment 7: Line 393: Say "additional amount" instead of "additionally amount"

Response 7: Following your comment, we revised the sentence (line 385).

Responses to Reviewer 3

Reviewer #3: I thank the authors for implementing my comments. However, I see room for further improvement on some points:

Response: Thank you very much for the positive words on our manuscript and your additional comments which were very helpful for improving our manuscript. We hope that you will be satisfied with the revised version in which we have incorporated your points. If you should have any further recommendations for improving our manuscript, please don´t hesitate to communicate these to us.

Comment 1: Thanks for your additional explanations on your theoretical model. However, I still miss a definition what a resource is and why it may buffer the effects of work demands on strain.

Response 1: Following your comment, we added a new paragraph into the introduction section where we now define personal, social and organisational resources. Furthermore, we added an example underlining that a resource may buffer the effect of work demands on strain. Please see line 94-104.

Comment 2: A note on the terms "stress and strain". In my opinion stress, like strain, is an individual reaction to an external demand or stressor (see Transactional Stress Model of Lazarus or Selyes adaptation syndrome). I therefore suggest using the term stressor instead of stress.

Response 2: Thank you very much for this suggestion. Following your comment, we changed the word stress into stressor (line 94, line 104).

Comment 3: All in all, it would be good to check once again that the main terms are used consistently. Especially workload and quantitative demands are used interchangeably.

Response 3: Thank you very much for this comment. We reviewed the whole manuscript for consistency in using the main terms, especially focusing on the terms ‘workload’ and ‘quantitative demands’ and made changes whenever needed. In this context, please see also our response to your comment no. 5 below.

Comment 4: It is not plausible why you only included those resources as moderators that showed statistically significant bivariate correlations with burnout, as this is not a prerequisite (neither statistical nor conceptual) for moderator analyses. Since your approach is explorative, I suggest that you use all resources of moderator analysis.

Response 4: Thank you very much for your comment. Due to the large amount of resource variables which were collected with our questionnaire, it was decided a priori “not to fish for significance” and to follow an a priori defined plan for statistical analysis which was coordinated with a biostatistician. This plan included that bivariate analyses should be performed to infer important variables for the regression-based moderation analysis (see line 189-190). However, to address your comment and in order not to miss important results, we computed moderator analyses with the resources family, religiosity/spirituality, gratitude of patients, recognition of patients/relatives and additional qualification in palliative care which were not significantly associated with burnout after bivariate testing. We found no moderation effects. 

Comment 5: Your explanation for not reporting interactions between specific demands in palliative care and resources is not very satisfying to me, as your study aims “to investigate the buffering role of resources on the relationship between workload and burnout among nurses in non-specialized palliative care settings, with consideration given to palliative care aspects, such as information on the ‘extent of palliative care’” (p.5) In this sense, your analyses do not completely meet the objective of your study.

Response 5: The present paper aimed to address the “quantitative demands”, because aspects such as time pressure and the quantitative amount of work, that need to be done within a certain amount of time, were identified as key job demands in the nursing profession (see for example Broetje, S., Jenny, G. J. and Bauer, G. F. 2020: The Key Job Demands and Resources of Nursing Staff: An Integrative Review of Reviews. Front Psychol. 2020; 11: 84. Published online 2020 Jan 31. doi: 10.3389/fpsyg.2020.00084). It is possible that “quantitative demands” will further increase in the future. We agree with you that we do not completely meet the objective of the study by using the word “workload” in the passage you cited from our manuscript. According to your comment, we could rename the title and the text passage you cited and use the term “quantitative demands” instead of “workload”. But we would reluctantly rename the title of the manuscript because workload is a much more common used term and thus, this title will reach more readers than the term “quantitative demands”. To make this aspect clear for the reader, we wrote in the Abstract, that the COPSOQ scale on ‘quantitative demands’ was used to measure workload (line 29-30). Furthermore, in the introduction section, we inform the reader that workload can be either qualitative (pertaining to the type of skills and/or effort needed in order to perform work tasks) or quantitative (the amount of work to be done and the speed at which it has to be performed) (line 75-77). However, following your comment, we incorporated some text in the manuscript (lines 110 and 152-153), aiming to highlight that we used the COPSOQ scale ‘quantitative demands’ to measure workload. Finally, please see the limitation section of the manuscript where your comment is addressed. In future studies, the different fields of nursing demands have to be carried out on the role of resources and this not only pertains for burnout, but also for other outcomes such as job satisfaction and health (line 388-391).

---

## [Decision Letter · Decision Letter 2]

23 Dec 2020

PONE-D-20-23720R2

The relationship between workload and burnout among nurses in non-specialized palliative care settings: the buffering role of personal, social and organisational resources

PLOS ONE

Dear Dr. Dietz,

Thank you for submitting your manuscript to PLOS ONE. After careful consideration, we feel that it has merit but does not fully meet PLOS ONE’s publication criteria as it currently stands. Therefore, we invite you to submit a revised version of the manuscript that addresses the remaining point raised during the review process.

We look forward to receiving your revised manuscript.

Kind regards,

Adrian Loerbroks

Academic Editor

PLOS ONE

Reviewers' comments:

Reviewer's Responses to Questions

**Comments to the Author**

1. If the authors have adequately addressed your comments raised in a previous round of review and you feel that this manuscript is now acceptable for publication, you may indicate that here to bypass the “Comments to the Author” section, enter your conflict of interest statement in the “Confidential to Editor” section, and submit your "Accept" recommendation.

Reviewer #3: (No Response)

2. Is the manuscript technically sound, and do the data support the conclusions?

Reviewer #3: No

3. Has the statistical analysis been performed appropriately and rigorously? 

Reviewer #3: Yes

4. Have the authors made all data underlying the findings in their manuscript fully available?

Reviewer #3: No

5. Is the manuscript presented in an intelligible fashion and written in standard English?

Reviewer #3: Yes

6. Review Comments to the Author

Reviewer #3: I thank the authors for implementing most of my previous comments.

However, I still find the argumentation of the article somehow inconsistent in one very central point: The title, the abstract and the study objective suggest that the study investigates the moderating effects of resources on the relationship between the specific demands of non-specialized palliative care and burnout. However, the study analyses the moderating effects of resources on the relationship between general demands in nursing (independently of specific demands of non-specialized palliative care) and burnout. I therefore suggest that you either report the interaction effects of between the demands in non-specialized palliative care and resources or remove the reference to palliative care from the title and clarify the abstract and the aim of the article accordingly.

7. PLOS authors have the option to publish the peer review history of their article (what does this mean?). If published, this will include your full peer review and any attached files.

Reviewer #3: No

---

## [Author Response · Author response to Decision Letter 2]

6 Jan 2021

Responses to Reviewer 3

Comment 1: I thank the authors for implementing most of my previous comments.

However, I still find the argumentation of the article somehow inconsistent in one very central point: The title, the abstract and the study objective suggest that the study investigates the moderating effects of resources on the relationship between the specific demands of non-specialized palliative care and burnout. However, the study analyses the moderating effects of resources on the relationship between general demands in nursing (independently of specific demands of non-specialized palliative care) and burnout. I therefore suggest that you either report the interaction effects of between the demands in non-specialized palliative care and resources or remove the reference to palliative care from the title and clarify the abstract and the aim of the article accordingly. 

Response 1: Thank you very much for the positive words on our manuscript and your final comment for improving it. We hope that we understood you right and incorporated your point to your satisfaction. 

As our title seems to be misleading, we removed the reference to palliative care from the title as you recommended. You were right, as you said that the study analyses the moderating effects of resources on the relationship between general demands in nursing and burnout. Accordingly, we revised the abstract (line 26-27, line 31-32). Additionally, we revised the aim of the study (line 110-112) and the discussion section (line 301).

---

## [Editor Report · Decision Letter 3]

8 Jan 2021

The relationship between workload and burnout among nurses: the buffering role of personal, social and organisational resources

PONE-D-20-23720R3

Dear Dr. Dietz,

We’re pleased to inform you that your manuscript has been judged scientifically suitable for publication and will be formally accepted for publication once it meets all outstanding technical requirements.

Kind regards,

Adrian Loerbroks

Academic Editor

PLOS ONE
---

## [Editor Report · Acceptance letter]

12 Jan 2021

PONE-D-20-23720R3 

The relationship between workload and burnout among nurses: the buffering role of personal, social and organisational resources 

Dear Dr. Dietz:

I'm pleased to inform you that your manuscript has been deemed suitable for publication in PLOS ONE. Congratulations! Your manuscript is now with our production department. 

Kind regards, 

on behalf of

Dr. Adrian Loerbroks 

Academic Editor

PLOS ONE